# Evaluation and Clinical Impact of Biofire FilmArray Pneumonia Panel Plus in ICU-Hospitalized COVID-19 Patients

**DOI:** 10.3390/diagnostics12123134

**Published:** 2022-12-12

**Authors:** Dolores Escudero, Jonathan Fernández-Suarez, Lorena Forcelledo, Salvador Balboa, Javier Fernández, Ivan Astola, Brigida Quindos, Rainer Campos, Fernando Vázquez, José Antonio Boga

**Affiliations:** 1Service of Intensive Medicine, Hospital Universitario Central de Asturias, 33011 Oviedo, Spain; 2Group of Translational Microbiology, Health Research Institute of Principado de Asturias (ISPA), 33011 Oviedo, Spain; 3Service of Microbiology, Hospital Universitario Central de Asturias, 33011 Oviedo, Spain; 4Department of Functional Biology, Universidad de Oviedo, 33006 Oviedo, Spain; 5Ophthalmology Institute Fernández-Vega, 33012 Oviedo, Spain

**Keywords:** ICU patients, respiratory infections, COVID-19, multiplex quantitative PCR systems

## Abstract

Microbiological diagnosis by using commercial multiplex quantitative PCR systems provides great advantages over the conventional culture. In this work, the Biofire FilmArray Pneumonia Panel Plus (FAPP+) was used to test 144 low respiratory tract samples from 105 COVID-19 patients admitted to an Intensive Care Unit (ICU), detecting 78 pathogens in 59 (41%) samples. The molecular panel was evaluated by using the conventional culture (CC) as comparator, which isolated 42 pathogens in 40 (27.7%) samples. The overall percentage of agreement was 82.6%. Values of sensitivity (93%), specificity (62%), positive predictive value (50%), and negative predictive value (96%) were obtained. The mean time elapsed from sample extraction to modification of antibiotic treatment was 7.6 h. A change in antimicrobial treatment after the FAPP+ results was performed in 27% of patients. The FAPP+ is a highly sensitive diagnostic method that can be used to significantly reduce diagnostic time and that allows an early optimization of antimicrobial treatment.

## 1. Introduction

The infection caused by betacoronavirus SARS-CoV-2, which is called COVID-19, can cause pneumonia and acute respiratory distress syndrome [1]. Within a few months, the disease spread worldwide, infecting millions of people and causing hundreds of thousands of deaths (https://coronavirus.jhu.edu/map.html, accessed on 15 October 2022), resulting in a major global health crisis.

Patients with bilateral COVID-19 bronchopneumonia admitted to Intensive Care Units (ICU) frequently suffer from bacterial, fungal, or viral superinfections. It is well known that a rapid identification of the causal pathogen allows early initiation of targeted antimicrobial treatment, reducing morbidity and mortality, hospital stay, and health care costs [2,3]. Microbiological identification by polymerase chain reaction (PCR) techniques with commercial multiplex quantitative PCR systems provides great advantages over the conventional culture. These faster techniques with high sensitivity and specificity can also estimate the bacterial load and provide information on antibiotic resistance determinants. In recent years, molecular microbiology has been changing the diagnostic approach to sepsis, meningitis, pneumonia, and other infectious pathologies [4,5,6,7,8,9,10,11,12,13,14] where rapid diagnosis is vital, especially in the case of critically ill patients.

In this study, respiratory bacterial co-infections in lower respiratory tract samples taken from ICU-hospitalized COVID-19 patients were evaluated by comparing a commercial multiplex qPCR system, the Biofire FilmArray Pneumonia panel Plus (FAPP+; BioFire Diagnostics, Salt Lake City, UT) with a conventional culture (CC). The impact of the use of the FAPP+ on the clinical management of ICU-hospitalized COVID-19 patients was assessed.

## 2. Materials and Methods

A prospective analysis of patients diagnosed with Covid-19 and admitted to an adult Intensive Care Unit (ICU) of a tertiary level university hospital from November 2020 to February 2021 was performed. The presence of SARS-CoV-2 was analyzed by detection of the viral genome using multiplex quantitative RT-PCR in nasopharyngeal exudates [15]. All patients in which a co-infection or superinfection was suspected were included in the study. The Clinical Pulmonary Infection Score (CPIS) and the 2016 IDSA guidelines were used for the diagnosis of ventilator-associated pneumonia (VAP) [16,17].

Tracheal aspirates, bronchoalveolar lavages, and sputum were collected from these patients and processed for conventional cultures (CC) and virological diagnosis and were tested by using the FAPP+. The conventional culture was performed according to the protocol recommended by the Spanish Society of Infectious Diseases and Clinical Microbiology [18]. For this purpose, they were plated on blood, chocolate, and McConkey agars (BioMérieux, Marcy l’Étoile, France) and incubated in a CO_2_ atmosphere for up to three days. The microorganisms grown in the culture were identified by MALDI TOF MS (Maldi Biotyper, Bruker Daltonics, Bremen, Germany). For virological diagnosis, samples were tested using diagnostic routine protocols of the Unit of Virology [19].

The FAPP+ is an automated multiplex PCR system, which contains the 27 most common pneumonia-producing pathogens (18 bacteria and 9 viruses). For bacteria, it uses a semiquantitative system to estimate the number of bacteria in the analyzed sample, with intervals of 10^4^, 10^5^, 10^6^ or ≥ 10^7^ copies of bacterial genome per milliliter of sample. Atypical bacteria, viruses, and resistance genes are reported only qualitatively. The pathogens included in the panel are: Acinetobacter calcoaceticus-baumannii complex, Enterobacter cloacae complex, *Escherichia coli, Haemophilus influenzae, Klebsiella aerogenes, Klebsiella oxytoca, Klebsiella pneumoniae group, Moraxella catarrhalis, Proteus spp., Pseudomonas aeruginosa, Serratia marcescens, Staphylococcus aureus, Streptococcus agalactiae, Streptococcus pneumoniae, Streptococcus pyogenes, Chlamydia pneumoniae, Legionella pneumophila, Mycoplasma pneumoniae,* adenovirus, coronavirus, human metapneumovirus, human rhinovirus/enterovirus, Middle East respiratory syndrome coronavirus, influenza A, influenza B, parainfluenza virus, and respiratory syncytial virus. The panel also includes seven antibiotic resistance genes encoding carbapenemases (blaIMP, blaKPC, blaNDM, blaOXA-48, blaVIM), extended spectrum beta-lactamases (blaCTX-M) and methicillin resistance (mecA/C and MREJ).

The FAPP+ results were compared to CC (considered as comparator method) and sensitivity, specificity, and positive and negative predictive values (PPV and NPV) for each analyte were determined. The two-sided 95% confidence intervals were calculated.

The results of the resistance genes detected by the FAPP+ were compared with the results of the susceptibility tests performed and interpreted according to EUCAST criteria. In all patients, the change of empirical therapy after the multiplex PCR panel was analyzed according to published guidelines [20].

Demographic data, days of admission, need for mechanical ventilation, ICU and in-hospital mortality, renal function analyses, procalcitonin and C-reactive protein levels, microbiological results, antibiotic treatment, modifications in treatment, and time from sample collection to treatment change were recorded.

## 3. Results

Overall, 144 respiratory samples belonging to 105 patients were analyzed in this study. Clinical characteristics of the enrolled patients are presented in Table 1. The distribution of the 144 respiratory samples was 138 (95.8%) tracheal aspirates, 5 (3.5%) bronchoalveolar lavages and one (0.7%) sputum. While 88 (61.1%) samples were collected at ICU admission, 56 (38.8%) were obtained during the ICU stays and collected when a superinfection was suspected.

The FAPP+ detected at least one potential pathogen in 59 (41.0%) of the 144 samples. Detection of more than one pathogen was observed in 13 samples, representing 9.0% of all samples and 22.0% of the FAPP+ positive samples. According to the CC, while 104 samples were negative or interpreted as normal airway microbiota, at least one pathogen was detected in 40 (27.7%) samples. Codetections were observed in two of them, representing 1,4% of all samples and 5% of the positive samples by CC. The overall percentage of agreement between both methods was 82.6%. The number and relative prevalence of each analyte detected by the FAPP+ and CC is shown in Table 2. Seventy-eight pathogens belonging to eleven bacterial species were detected by the FAPP+, the most prevalent being *H. influenzae*, *S. aureus*, *P. aeruginosa*, and *K. pneumoniae* which were found in 25 (17.4%), 16 (11.1%), 10 (6.9%), and 6 (4.2%) samples, respectively. All other targets were detected in 5 (6,4%%) or fewer of the samples. Thirty-nine of the 42 pathogens detected by CC were also detected by the FAPP+. Three *S. pneumoniae* were only detected by CC. In addition, there were 3 *Raoutella* sp. and 1 *Bordetella sp*. reported from cultures that are not targeted by the FAPP+. Sensitivity, specificity, PPV, and NPV were calculated with respect to the comparator method of bacterial culture (Table 2).

The distribution of the pathogens identified in the samples collected from patients at the moment of ICU admission and those obtained from patients already admitted to the ICU and collected when superinfection was suspected is shown in Table 3. While *H. influenzae*, *S. aureus,* and *S. pneumoniae* were the pathogens mainly identified at the moment of ICU admission, an increase in *P. aeruginosa* and several enterobacterales, such as *E. coli*, *E. cloacae*, *Klebsiella* sp., and *S. marcescens*, were detected in patients with superinfection during ICU admission. The FAPP+ detected 10 pathogens more than CC (30 vs. 20, increasing the diagnostic yield by 66.7%) in the samples collected at the moment of ICU admission and 26 pathogens more in those obtained from patients already admitted to the ICU (48 vs. 22, increasing the diagnostic yield by 118.2%).

The comparison between the bacterial counts detected by culture and those reported by the FAPP+ is shown in Table 4. It is noteworthy that fifteen CC negative samples had a bacterial load higher than or equal to 10^6^ copies/mL.

The detection of antibiotic resistance markers by the FAPP+ indicated the presence of 2 *K. pneumoniae* ESBL and/or carbapenemase and a single *S. aureus* MR. These results were concordant with the phenotypes obtained in the antibiogram.

Twenty-six (24.7%) patients tested at the moment of admission had received previous antibiotic treatment, as had 49 (87.5%) of those which were tested during ICU admission. A change of empirical antimicrobial treatment was performed in 39 (27%) patients, with escalation or initiation of treatment in 25 (64.1%) cases and de-escalation in 14 (35.9%) cases. Therapeutic optimization was performed in 17 (19.3%) of the patients analyzed at the moment of admission and in 22 (37.5%) of those suspected of superinfection, with an escalation of 58.8% and 68.2%, and a de-escalation of 41.2% and 31.8%, respectively.

The mean time elapsed from sample extraction to modification of antibiotic treatment in patients diagnosed by the FAPP+ was 7.6 h (10.8 h in the case of newly admitted patients and 5.4 h in those who were infected during the ICU stay). The results of the CC, which were obtained after the results of the FAPP+, led to modification of antimicrobial treatment in 18 cases (12.5%).

## 4. Discussion

In COVID-19 patients hospitalized in the ICU, respiratory superinfections are frequent and increase with the number of days of mechanical ventilation [21,22,23,24,25,26]. In clinical practice, broad-spectrum antibiotic empiric therapy is initiated while waiting for the microbiological results. Nevertheless, some studies have shown that in hospitalized patients with pneumonia, etiological diagnosis using traditional methods is only achieved in 38% of cases [27]. These poor results support the use of other diagnostic approaches. Among these, the use of syndromic PCR panels, including the most frequent agents causing lower respiratory tract infections and their most important antibiotic resistance markers, offer a faster diagnosis and consequently a potentially prompt optimization of antibiotic treatment [28,29,30,31,32]. According to this fact, respiratory samples from COVID-19 patients hospitalized in our ICU were tested by the syndromic panel FAPP+ and evaluated using conventional culture as comparator method. As has been described by other authors, a relatively high number of cases of superinfections among these patients were found, of which *P. aeruginosa, H. influenzae, S. pneumoniae*, and *S. aureus* were the most frequently detected pathogens [29,33,34,35]. In our study, the positive rates of the FAPP+ and the CC were 41.7% vs. 27.7%, respectively. Slightly higher positive rates were reported by a recent meta-analysis of seven studies performed in patients with COVID-19 [28]. A study evaluating the diagnostic performance of a similar syndromic panel in cases of community-acquired pneumonia confirmed that this panel detected almost double the number of potential bacterial pathogens than did a package of various pneumonia tests, including cultures, antigen detection, and PCR testing [36]. According to the diagnostic performance data of FAPP+ using CC as the comparator method, a sensitivity of 100%, except for *S. pneumoniae*, and a specificity of 98–100%, except for *H. Influenzae*, were calculated. These data are concordant with those obtained by other authors [37,38]. The PPV and NPV values found are similar to those described in previous studies [35]. The low sensitivity for *S. pneumonia* (51.7%) can be explained by the detection of three cases of *S. pneumoniae* only by CC. A possible explanation is the presence of mutations in the PCR region target, a phenomenon already known in Gram-negative bacteria, such as has been reported in other studies [35]. The number of samples in which more than one pathogen was recovered was higher when the molecular panel was used (22% vs. 5%). These data suggest a higher sensitivity of the panel and a not excessively high frequency of polymicrobial infection. Almost 40% of the samples that were positive according to the FAPP+ and negative through CC had a quantification higher than 106, which is considered significant. While microbiological count obtained by CC is measured by CFU/mL, these data by syndromic panel are measured as DNA copies per mL, and it is important to highlight that the latter results are not affected by previous antibiotic treatment. A possible explanation for the presence of discordant results (FAPP+ positive and CC negative) is the use of previous antibiotics, which are well known to inhibit bacterial growth in cultures. Molecular methods can detect non-viable microorganisms resulting in a higher sensitivity compared to conventional cultures, as has been reported in patients with community-acquired pneumonia [39]. Although a high sensitivity is an advantage, in clinical practice that could generate discrepancies between results obtained by various techniques. Quantitative systems can help to differentiate colonization from infection, since isolates with higher bacterial load are more likely to be clinically significant as has been reported in previous studies [38]. Interpreting whether the pathogens detected are clinically relevant or simply colonizing represents a challenge that requires a clinical and epidemiological assessment, a diagnostic stewardship approach, and a good exchange of information between clinician and microbiologist. The sensitivity, specificity, PPV, and NPV values of a new method are calculated by comparison to a method considered the gold standard, which is the conventional culture in the case of syndromic panels. Nevertheless, the limitations of culture and its low diagnostic yield in respiratory samples [27,39], especially in patients who had previous antimicrobial treatment, support the use of another reference standard.

Rapid diagnosis and early treatment are cornerstones in the management of critically ill patients and can contribute to reducing morbidity and mortality, hospital stay, antibiotic resistance, and healthcare costs [2,37]. Unfortunately, and as a limitation, the present study was not designed to assess the impact of the FAPP+ application in the reduction of hospital length of stay and mortality of the patients included; however, it is well known that, especially in infections such as sepsis or severe pneumonia, a diagnostic or therapeutic delay negatively influences the patient’s evolution. Moreover, previous studies have demonstrated that regardless of the source of infection, its severity and origin (nosocomial or community), the delay and the inadequate antibiotic treatment onset increases mortality and hospital stays [40].

In our study, the use of the multiplex-PCR system allowed an improvement of antibiotic treatment by escalation or de-escalation in 27% of patients. A similar result with a higher range (34–37%) was recently reported [28]. Another advantage of the use of the FAPP+ was the significant decrease in the mean time between the extraction of the sample and the instauration of a targeted antibiotherapy (7.6 h vs. 24–72 h, using microbiological identification and antibiotic susceptibility testing after conventional cultures). In this sense, the rapid detection of resistance genes by the FAPP+, which was in all cases consistent with those obtained phenotypically by conventional methods, is also of great clinical relevance, since it allows early initiation of targeted antibiotic treatment. Furthermore, although the FAPP+ does not predict susceptibility/resistance to all antibiotics, the identification of the microorganism together with knowledge of the local epidemiology and resistance patterns can help to establish prompt semitargeted antimicrobial therapy, reducing the empiricism with which respiratory bacterial infections are treated initially [41].

Among the limitations of the FAPP+ are the non-inclusion of some clinically relevant pneumonia-causing pathogens (bacteria and fungi) and the detection of a limited number of resistance genes. On the other hand, its relatively high price demands optimization of the cost/benefit ratio by protocolizing its indication with the help of the Microbiology Service. This study has some limitations: it was carried out in a single center and with a small series, which, as previously noted, did not allow us to analyze the impact on antibiotic consumption, average length of hospital stay and mortality. Although several viral pathogens are included in the panel FAPP+, this type of infection was not detected in our patients, which supports the absence of seasonal respiratory viruses during the studied period (second wave of COVID-19 pandemic), but which prevented us from evaluating the diagnostic performance of the detection of these pathogens.

## 5. Conclusions

This study supports the conclusion that the FAPP+ offers a rapid diagnosis of respiratory bacterial infections with a high sensitivity and specificity. The molecular approach cannot completely replace conventional cultures since the latter continue to be the gold standard and allow the isolation of the microorganism for further antimicrobial susceptibility testing. However, the FAPP+ can be a complement for the early management of pneumonia, which proved to be an excellent tool to rapidly identify etiological agents, to guide clinical decisions early, and to optimize the use of antimicrobials, especially in the context of diagnostic and antimicrobial stewardship initiatives.

## Figures and Tables

**Table 1 diagnostics-12-03134-t001:** Clinical characteristics of patients.

Variables	Patients (*n* = 105)
Male Sex	79 (75.2%)
Age (years)	64 (30–80)
Hospitalization days	31 (8–136)
ICU stay (days)	21 (3–102)
Hospital mortality	1 (0.9%)
ICU mortality	21 (23.8%)
Mechanical ventilation	105 (99.3%)
Leukocytosis	65 (45.1%)
Elevated procalcitonin levels	27 (18.7%)
Elevated CRP	64 (44.4%)
Impaired kidney functions	36 (12.5%)

Data are presented as: *n*(%)—Median [IQR].

**Table 2 diagnostics-12-03134-t002:** Performance summary of the FAPP+ versus those of the conventional culture.

						Sensitivity		Specificity		PPV		NPV	
	No. of Pathogens					TP/(TP + FN)		TN/(TN + FP)		TP/(TP + FP)		TN/(TN + FN)	
Pathogen	Total	TP	FN	FP	TN	%	95% CI	%	95% CI	%	95% CI	%	95% CI
*H. influenzae*	25	8	-	17	119	100	67.6–100	87.5	80.9–92.0	32	17.2–51.6	100	96.9–100
*S. aureus*	16	11	-	5	128	100	74.1–100	96.2	91.5–98.4	68.8	44.4–85.8	100	97.1–100
*P. aeruginosa*	10	7	-	3	134	100	64.6–100	97.8	93.8–99.3	70	39.7–89.2	100	97.2–100
*S. pneumoniae*	8	4	3	1	136	57.1	25.0–84.2	99.3	95.4–99.9	80	37.6–96.4	97.5	93–99.2
*K. pneumoniae*	6	3	-	3	138	100	43.9–100	97.9	93.9–99.3	50	18.8–81.2	100	97.3–100
*E. coli*	5	2	-	3	139	100	34.2–100	97.9	94–99.3	40	11.8–76.9	100	97.3–100
*E. cloacae*	3	3	-	-	141	100	43.9–100	100	97.3–100	100	43.9–100	100	97.3–100
*S. marcescens*	2	1	-	1	142	100	20.7–100	99.3	96.1–99.9	50	9.5–90.5	100	97.4–100
*S. agalactiae*	2	-	-	2	142	--	--	98.6	95.1–99.6	0	0–65.8	100	97.4–100
*K. aerogenes*	2	-	-	2	142	--	--	98.6	95.1–99.6	0	0–65.8	100	97.4–100
*M. catarrhalis*	2	-	-	2	142	--	--	98.6	95.1–99.6	0	0–65.8	100	97.4–100
Total	81	39	3	39	63	92.9	81.0–97.5	61.8	52.1–70.6	50	39.2–60.8	95.5	87.5–98.4

PPV: Positive predictive value, NPV: Negative predictive value, TP: True positive (FAPP+ and CC positive), FP: False positive (FAPP+ positive and CC negative), FN: False negative (FAPP+ negative and CC positive), FN: False negative (FAPP+ and CC negative).

**Table 3 diagnostics-12-03134-t003:** Distribution of pathogens detected at the moment of admission and in cases of superinfection.

Pathogen	Admission	Superinfection	Total
*H. influenzae*	13	12	25
*S. aureus*	7	9	16
*P. aeruginosa*	-	10	10
*S. pneumoniae*	4	1	8
*K. pneumoniae*	-	6	6
*E. coli*	1	4	5
*E. cloacae*	1	2	3
*S. marcescens*	-	2	2
*S. agalactiae*	2	-	2
*K. aerogenes*	-	2	2
*M. catarrhalis*	2	-	2
TOTAL	30	48	78

**Table 4 diagnostics-12-03134-t004:** Comparison of the microbiological count of pathogens detected by the FAPP+ and CC.

CC Count (fpu/mL)	FAPP+ Count (Copies/mL)				
	≥10^7^	10^6^	10^5^	10^4^	No Detected
>10^6^	19	2	-	-	2
10^6^	3	4	-	-	1
10^5^	3	5	3	-	-
No detected	8	7	15	9	-

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
