# Peer review of "Evaluation and Clinical Impact of Biofire FilmArray Pneumonia Panel Plus in ICU-Hospitalized COVID-19 Patients"

_diagnostics, 2022, doi:10.3390/diagnostics12123134_

Round 1

Reviewer 1 Report

It has been suggested that microbiological diagnosis by using commercial multiplex quantitative PCR systems including the Biofire FilmArray Pneumonia Panel Plus (FAPP+)provides great advantages over conventional culture (CC). The authors have examined that 144 low respiratory tract samples from 105 COVID-19 patients admitted to an Intensive Care Unit (ICU). They found that the FAPP+ system detected 78 pathogens in 59 (41%) samples, while the molecular panel was evaluated by using conventional culture (CC) as comparator, which isolated 42 pathogens in 40 (27.7%) samples. In the parallel examinations, the overall percentage of agreement was 82.6%. It has a high sensitivity (93%), and the modest specificity (62%).

Seventy-eight pathogens belonging to eleven bacterial species were detected by the FAPP+ being the most prevalent H. influenzae, S. aureus, P. aeruginosa, and K. pneumoniae which were found in 25

(17.4%), 16 (11.1%), 10 (6.9%), and 6 (4.2%) samples, respectively.

Theses pathogens are clinically reasonable and could be detected in patients with COVID-19 in ICU settings.

It is reasonable to assume that FAPP+ is a highly sensitive diagnostic method that can be used to reduce significantly diagnostic time and allows an early optimization of antimicrobial treatment.

However, the specificity of the FAPP+ is reasonable, but not high.

Thus, the parallel culture studies are needed to confirm the rapid diagnosis by FAPP+ .  

Author Response

Reviewer 1 response:

It has been suggested that microbiological diagnosis by using commercial multiplex quantitative PCR systems including the Biofire FilmArray Pneumonia Panel Plus (FAPP+)provides great advantages over conventional culture (CC). The authors have examined that 144 low respiratory tract samples from 105 COVID-19 patients admitted to an Intensive Care Unit (ICU). They found that the FAPP+ system detected 78 pathogens in 59 (41%) samples, while the molecular panel was evaluated by using conventional culture (CC) as comparator, which isolated 42 pathogens in 40 (27.7%) samples. In the parallel examinations, the overall percentage of agreement was 82.6%. It has a high sensitivity (93%), and the modest specificity (62%).

Seventy-eight pathogens belonging to eleven bacterial species were detected by the FAPP+ being the most prevalent H. influenzae, S. aureus, P. aeruginosa, and K. pneumoniae which were found in 25

(17.4%), 16 (11.1%), 10 (6.9%), and 6 (4.2%) samples, respectively.

Theses pathogens are clinically reasonable and could be detected in patients with COVID-19 in ICU settings.

It is reasonable to assume that FAPP+ is a highly sensitive diagnostic method that can be used to reduce significantly diagnostic time and allows an early optimization of antimicrobial treatment.

However, the specificity of the FAPP+ is reasonable, but not high.

Thus, the parallel culture studies are needed to confirm the rapid diagnosis by FAPP+

We appreciate the reviewer comments and the article valuation. We agree with him/her that FAPP+ should not replace conventional cultures however it can complement it in the diagnosis workflow. This tests has the advantage of speed and high sensitivity, but culture continues to be the gold standard and the technique that allows the isolation of microorganisms for subsequent performance of antibiotic susceptibility testing. We have modified the conclusions in the new version of the manuscript in order to clarify this point.

Reviewer 2 Report

General Comments:

        It is important to develop new technics to early detect microbials in critically ill patients since early correct antibiotic use help patients survive. Quantitative PCR system is one of these methods. However, this study did not point out the issue which critical care physicians wants to know. That is whether are there benefits in mortality or ICU stay after using this commercial multiplex PCR system. Authors should analyze survival benefit in patients between presence or absence of early modification of antibiotic.

Specific Comments:

1.    In Methods section (Line 55-56), the definition of VAP should be updated.

2.    In Table 1, how were hospital and ICU mortality calculated? Why was hospital mortality rate lower than ICU mortality rate?

3.    Abbreviation in Table 2 should be stated.

4.    In Line 131-136 (Results section), this paragraph is poorly understood and needs rewriting. 

5.    In Line 178 (Discussion section), does NPV miswrite to PNV?

6.    Because PCR system cannot provide correct antibiotic sensitivity result, it is not appropriate to conclude that FAPP+ is an excellent tool to early guide clinical decisions and optimizes the use of antimicrobials in Conclusion section.

Author Response

Reviewer 2 response:

General Comments:

        It is important to develop new technics to early detect microbials in critically ill patients since early correct antibiotic use help patients survive. Quantitative PCR system is one of these methods. However, this study did not point out the issue which critical care physicians wants to know. That is whether are there benefits in mortality or ICU stay after using this commercial multiplex PCR system. Authors should analyze survival benefit in patients between presence or absence of early modification of antibiotic.

We thank the reviewer for his/her comments. Assessing the reduction in length of stays, costs and mortality associated with an intervention in the healthcare environment is complex. Unfortunately, our study was not designed for this purpose and the number of patients evaluated may be too small to draw conclusions in this regard. However, it is well known that some infections such as sepsis or severe pneumonia are considered time-dependent diseases, such as stroke or acute myocardial infarction, in which diagnostic or therapeutic delay negatively influences the patient's evolution. Some studies have shown that regardless of the source of infection, its severity and its origin (nosocomial or community), the delay and the inadequate antibiotic treatment onset increases mortality and the hospital stays (Kollef MH et al. Timing of antibiotic therapy in the ICU. Crit Care. 2021 Oct 15;25:360) We discussed about this in the new version of the manuscript and we have added as a limitation of the study not having analyzed the benefits of the application of this technique in terms of reduction of hospital stays and mortality.

Specific Comments:

  1. In Methods section (Line 55-56), the definition of VAP should be updated.

VAP has been definited according to the 2016 IDSA guidelines

  1. In Table 1, how were hospital and ICU mortality calculated? Why was hospital mortality rate lower than ICU mortality rate?

While ICU mortality refers to patients who die while in the ICU, hospital mortality refers to those who die on the ward and who had previously been discharged from the ICU.

  1. Abbreviation in Table 2 should be stated.

Abbreviation in Table 2 has been stated.

  1. In Line 131-136 (Results section), this paragraph is poorly understood and needs rewriting. 

According to manufacturer´s instructions, semi-quantitative Bin (copies/mL) results generated by the FilmArray Panel are not equivalent to CFU/mL and do not consistently correlate with the quantity of bacterial analytes compared to CFU/mL. Nevertheless, it should be noted that negative samples by CC had values of copies/ml higher than 106 by FAPP+. The paragraph has been rewritten to emphasize this fact.

  1. In Line 178 (Discussion section), does NPV miswrite to PNV?

PNV has been changed by NPV

  1. Because PCR system cannot provide correct antibiotic sensitivity result, it is not appropriate to conclude that FAPP+ is an excellent tool to early guide clinical decisions and optimizes the use of antimicrobials in Conclusion section.

We have introduced a paragraph in the discussion and modified the conclusion to better explain this idea. Cultures are still needed for antimicrobial susceptibility testing, however although the FAPP+ does not predict susceptibility/resistance to all antibiotics, the identification of the microorganism together with knowledge of the local epidemiology and resistance patterns can help to establish prompt semitargeted antimicrobial therapy reducing the empiricism with which respiratory bacterial infections are treated initially.

Round 2

Reviewer 2 Report

No further comments.